Development and validation of a nomogram risk prediction model for malignancy in dermatomyositis patients: a retrospective study

Zhong Jiaojiao 1 2
He Yunan 3
Ma Jianchi 1
Lu Siyao 1
Wu Yushi 1
Zhang Junmin zhjunm@mail.sysu.edu.cn 1
1 Department of Dermatology, Sun Yat-sen Memorial Hospital, Sun Yat-sen University , Guangzhou , Guangdong , China
2 Institute of Dermatology, Chinese Academy of Medical Sciences and Peking Union Medical College , Nanjing , China
3 Reproductive Medicine Center, Tangdu Hospital, Air Force Medical University , Xi’an , China
Sergi Consolato
Electronic publication date: 2021 Dec 9
Publication date: 2021
Volume: 9
Electronic Location ID: e12626
Received 2021 Jul 23; Accepted 2021 Nov 19
Copyright: ©2021 Zhong et al.
Copyright year: 2021
Copyright holder: Zhong et al.
License: This is an open access article distributed under the terms of the Creative Commons Attribution License, which permits unrestricted use, distribution, reproduction and adaptation in any medium and for any purpose provided that it is properly attributed. For attribution, the original author(s), title, publication source (PeerJ) and either DOI or URL of the article must be cited.
License URL: https://creativecommons.org/licenses/by/4.0/

Keywords: Dermatomyositis, Malignancy, Predictor, Nomogram, Risk prediction model

Funding: The authors received no funding for this work.

==============================
Background

Dermatomyositis accompanied with malignancy is a common poor prognostic factor of dermatomyositis. Thus, the early prediction of the risk of malignancy in patients with dermatomyositis can significantly improve the prognosis of patients. However, the identification of antibodies related to malignancy in dermatomyositis patients has not been widely implemented in clinical practice. Herein, we established a predictive nomogram model for the diagnosis of dermatomyositis associated with malignancy.

Methods

We retrospectively analyzed 240 cases of dermatomyositis patients admitted to Sun Yat-sen Memorial Hospital, Sun Yat-sen University from January 2002 to December 2019. According to the year of admission, the first 70% of the patients were used to establish a training cohort, and the remaining 30% were assigned to the validation cohort. Univariate analysis was performed on all variables, and statistically relevant variables were further included in a multivariate logistic regression analysis to screen for independent predictors. Finally, a nomogram was constructed based on these independent predictors. Bootstrap repeated sampling calculation C-index was used to evaluate the model’s calibration, and area under the curve (AUC) was used to evaluate the model discrimination ability.

Results

Multivariate logistic analysis showed that patients older than 50-year-old, dysphagia, refractory itching, and elevated creatine kinase were independent risk factors for dermatomyositis associated with malignancy, while interstitial lung disease was a protective factor. Based on this, we constructed a nomogram using the above-mentioned five factors. The C-index was 0.780 (95% CI [0.690–0.870]) in the training cohort and 0.756 (95% CI [0.618–0.893]) in the validation cohort, while the AUC value was 0.756 (95% CI [0.600–0.833]). Taken together, our nomogram showed good calibration and was effective in predicting which dermatomyositis patients were at a higher risk of developing malignant tumors.

Introduction

Dermatomyositis is a group of autoimmune diseases that mainly damage the skin and muscles and is often accompanied by tumors and internal organ damage. Skin lesions are characterized by periorbital erythema, Gottron’s sign, poikiloderma and V-neck sign. Myositis often manifests as proximal muscle weakness, joint pain and dysphagia. The relationship between dermatomyositis and malignancy was first reported by Stertz (1916). As a paraneoplastic syndrome, tumors commonly associate with dermatomyositis, including nasopharyngeal cancer, breast cancer, thyroid cancer, ovarian cancer and lung cancer and so on. The pathogenesis is reportedly related to the immune cross-reaction between tumors and normal host tissues or the secretion of functional peptides and hormones (Pelosof & Gerber, 2010). Dermatomyositis patients with underlying malignancy have a poorer prognosis compared to those without. It has been reported that the incidence of malignancy in adults with dermatomyositis ranges from 23% to 45% (Maoz et al., 1998; Fang et al., 2016; Requena et al., 2014). The 5-year survival rate of dermatomyositis patients with malignancy ranges from 51.8% to 74.2% (Fang et al., 2016; Kim et al., 2011). In addition, some studies have reported that the incidence of malignancy in dermatomyositis is highest in the first year after diagnosis (Chen et al., 2014). Therefore, it is critical to predict the probable risk of malignancy in dermatomyositis patients early.

Recently, it has been reported that a variety of myositis-specific autoantibodies (MSAs) associated with tumors has been found in patients with dermatomyositis, including transcriptional intermediary factor 1-gamma (TIF-1γ), nuclear matrix protein (NXP-2), anti-small ubiquitin-like modifier activating enzyme (SAE), anti-3-hydroxy-3-methylglutaryl-coenzyme A reductase (HMGCR), etc (Trallero-Araguas et al., 2010). Among them, TIF1γ has the highest positive rate in dermatomyositis patients with underlying malignancy (Hida et al., 2016), with about 60% to 80% of positive TIF-1 patients harboring a malignancy (Trallero-Araguas et al., 2010). Although TIF1γ shows a strong connection between malignancy and myositis, many hospitals have not carried out TIF-γ1 testing clinically due to the high cost. In addition, a previous meta-analysis (Lu et al., 2014) has shown that male dermatomyositis patients older than 45-year-old with dysphagia, cutaneous necrosis, cutaneous vasculitis, rapid onset of myositis (<4 weeks), elevated creatine kinase (CK), elevated erythrocyte sedimentation rate (ESR), and elevated C-reactive protein (CRP) were more likely to develop malignancy, while interstitial lung disease (ILD), arthritis, Raynaud syndrome and the presence of anti-Jo-1 decreased that risk. However, to date, no quantitative predictive model based on the above predictors has been established to help identify malignancy incidence in patients with dermatomyositis.

Therefore, to that end, we established a clinical prediction model for diagnosing dermatomyositis with underlying malignancy. Nomograms are generally used to transform complex regression equations into visual graphics, making the predictive model more readable and convenient for patient evaluation. A score is assigned according to each influencing factor’s degree of contribution to the outcome of the variable. After which, the scores are added together to obtain the total score. Finally, the predicted value of the individual outcome event is calculated through the function conversion relationship between the total score and the probability of an outcome event (Balachandran et al., 2015; Iasonos et al., 2008). Our predictive model was further verified both internally and externally to ensure good clinical reproducibility. To our knowledge, this is the first nomogram constructed to predict the probability of malignancy in dermatomyositis patients.

Materials & Methods

Patients

Since this study was a retrospective study, all the patients included in the study had been discharged from the hospital, and the signature of written consent from the patients could not be obtained. However, to obtain the patients’ consent, the researchers have contacted the patients in the study one by one by telephone and informed them of the contents of the informed consent forms. Oral consent has been obtained, and the patients have authorized us to use the data during their hospital stay in the study. The Ethics Committee of Sun Yat-sen Memorial Hospital, Sun Yat-sen University, has approved the oral consent for this study instead of the written informed consent (SYSEC-KY-KS-2020-038).

It was registered in the Chinese Clinical Trial Registry (Registration No. ChiCTR2000031286). This study followed all relevant regulations in terms of data collection and storage. The clinical data of patients diagnosed with dermatomyositis at Sun Yat-sen Memorial Hospital, Sun Yat-sen University, from January 2002 to December 2019 were collected from the hospital database.

The inclusion criteria were according to the diagnosis criteria of dermatomyositis proposed by Bohan & Peter (1975): (1) proximal symmetry of weak muscles in the extremities, with or without difficulty in swallowing and respiratory muscle weakness; (2) elevated muscle enzymes; (3) electromyography showing myogenic changes; (4) abnormal muscle biopsy; (5) characteristic cutaneous changes.

Exclusion criteria were as follows: (1) incomplete clinical data records and inadequate examinations; (2) unconfirmed diagnosis of dermatomyositis; (3) malignancy occurred before the diagnosis of dermatomyositis; (4) patients with overlapping syndrome, that is, combined other rheumatic immune diseases (such as systemic lupus erythematosus, scleroderma, polymyositis, rheumatoid arthritis, etc.).

The patients were selected according to the inclusion and exclusion criteria. After a diagnosis of dermatomyositis, they were divided into the malignancy and non-malignancy groups. Depending on the year of the first admission, patients from 2002 to 2015 were designated into the training cohort, while patients from 2016 to 2019 were assigned to the validation cohort.

Data collection

The following clinical data were collected: (1) general information of the patient: age, gender; (2) clinical manifestations involving Gottron’s sign, periungual erythema, poikiloderma, refractory itching, V-neck sign, periorbital erythema, Raynaud’s phenomenon, joint pain, proximal muscle weakness, and dysphagia; (3) laboratory indicators: antinuclear antibodies (ANA), anti-Jo-1 antibodies, CK, lactate dehydrogenase (LDH), carbohydrate antigen 125 (CA125), carbohydrate antigen 19-9 (CA19-9); (4) complications: malignancy, ILD, respiratory failure.

Variables definition

Dermatomyositis patients with associated malignancy were diagnosed by pathological biopsy, including biopsy after tumor resection, endoscopic biopsy, and lymph node biopsy . ILD was diagnosed by chest X-ray or high-resolution CT (characterized by an increased lung density with a frosted glass-like appearance in bilateral lower lung fields), pulmonary function testing, and lung biopsy. Respiratory failure refers to severe obstruction of pulmonary ventilation and/or ventilation function caused by various reasons, leading to hypoxia with/without carbon dioxide retention, with arterial blood gas values of PaO2 below 60 mmHg, with or without a PaCO2 value above 50 mmHg.

Model development

According to the normal data range, the continuous variables in the predictors were converted into binary variables. Then, SPSS ver. 22.0 (IBM Co., Armonk, NY, USA) was used to compare the composition of the basic data in the training and validation cohort, using the χ2 test (or Fisher’s exact test) and t-test (or Mann–Whitney U test) for categorical and continuous variables, respectively. Univariate logistic regression was used to evaluate the relationship between each independent variable and the occurrence of malignancy. The variables with significant differences (P < 0.1) were further incorporated into our multivariate logistic analysis to screen for independent predictors (Collins et al., 2015). The selected predictors were introduced into R ver. 3.1.2 (R Foundation for Statistical Computing, Vienna, Austria; http://www.r-project.org/), and a nomogram prediction model was constructed using the rms software package.

Model validation

Internal validation was performed using the bootstrap method for repeated sampling (1,000 times). The calibration of the nomogram was evaluated by the Concordance index (C-index). The calibration curve was analyzed by plotting the predicted nomogram and the actual probability of malignancy in patients with dermatomyositis. The C-index of the calibration curve ranged from 0.5 to 1. The closer it is to 1, the more accurate the model’s prediction results are in accordance with the actual situation. For external validation, dermatomyositis patients from January 2016 to December 2019 were selected according to the same inclusion and exclusion criteria. The receiver operating characteristic curve (ROC) was drawn, and the area under the curve (AUC) was calculated to evaluate the model’s discrimination ability. The closer the AUC value is to 1, the better the model’s discrimination is.

Results

Patients characteristic

A total of 289 patients with dermatomyositis were enrolled in this study. After excluding patients with incomplete data, malignancy occurring before dermatomyositis, and patients with other rheumatoid immune diseases, a total of 240 cases were selected for further analysis, including 93 men and 147 women. The average age of the patients was 46.99 ± 18.17 years. Among them, 54 cases had malignancy. The top three malignant tumors were nasopharyngeal cancer (37.0%), lung cancer (16.7%), breast cancer (13.0%) Table S1. All eligible patients were grouped by the year of admission, 168 patients admitted from 2002 to 2015 were selected for our training cohort, and 72 patients admitted to the hospital from 2016 to 2019 were picked for our validation cohort (Fig. 1). The ratio of the two groups was 7:3. The demographic, clinical characteristics and experimental results of the training and validation cohorts were similar (Table 1).

Table 1 Characteristics of patients with dermatomyositis.

Characteristic	Training cohort (n = 168)	Validation cohort (n = 72)		
	N	Percent (%)	N	Percent (%)	P	
Basic information							
Age (years), mean ± SD	48.09 ± 18.69	44.43 ± 16.73	0.153	
Sex					0.977	
Male	65	38.7	28	38.9		
Female	103	61.3	44	61.1		
Clinical manifestation							
Gottron’s sign					0.167	
yes	84	50.0	43	59.7		
no	84	50.0	29	40.3		
Periungual erythema					0.792	
yes	19	11.3	9	12.5		
no	149	88.7	63	87.5		
Poikiloderma					0.019	
yes	47	28.0	10	13.9		
no	121	72.0	62	86.1		
Refractory itching					0.562	
yes	21	12.5	11	15.3		
no	147	87.5	61	84.7		
V-neck sign					0.842	
yes	70	41.7	31	43.1		
no	98	58.3	41	56.9		
Periorbital erythema					0.185	
yes	113	67.3	42	58.3		
no	55	32.7	30	41.7		
Raynaud’s phenomenon					0.778	
yes	8	4.8	4	5.6		
no	160	95.2	67	94.4		
Joint pain					0.258	
yes	23	13.7	14	19.4		
no	145	86.3	58	80.6		
Proximal muscle weakness					0.829	
yes	119	70.8	50	69.4		
no	49	29.2	22	30.6		
Dysphagia					0.837	
yes	37	22.0	15	20.8		
no	131	78.0	57	79.2		
Complication						
Malignant tumor					0.157	
yes	42	25.0	12	16.7		
no	126	75.5	60	83.3		
Interstitial pneumonia					0.411	
yes	72	42.9	35	48.6		
no	96	57.1	37	51.4		
Respiration failure					0.205	
yes	30	17.9	18	25.0		
no	138	82.1	54	75.0		
Laboratory values						
CK (U/L)					0.058	
≥198	69	41.1	20	27.8		
<198	99	58.9	52	72.2		
LDH (U/L)					0.104	
≥300	103	61.3	36	50.0		
<300	65	38.7	36	50.0		
ANA					0.862	
Positive	103	61.3	45	62.5		
Negative	65	38.7	27	37.5		
Anti-Jo-1					0.004	
Positive	4	2.4	8	11.1		
Negative	164	97.6	64	88.9		
CA125 (U/ml)					0.375	
≥35	15	8.9	4	5.6		
<35	153	91.1	68	94.4		
CA19-9(U/ml)					0.200	
≥37	21	12.6	5	6.9		
<37	146	87.4	67	93.1		
Notes.

CK creatine kinase

LDH lactate dehydrogenase

ANA antinuclear antibody

CA125 carbohydrate antigen 125

CA19-9 carbohydrate antigen 19-9

Figure 1 Flow chart for cases selection.

Predictive factors for dermatomyositis patients with malignancy

In the training cohort, univariate logistic regression was used to identify potential predictors of dermatomyositis with malignancy (Table 2). When contrasting the malignant tumor group to the non-malignant tumor group, we found that patients with dysphagia (odds ratio (OR) = 4.224; 95% CI [1.932–9.233]; P < 0.001), male gender (OR = 1.864; 95% CI [0.909–3,780]; P = 0.084), poikiloderma (OR = 2.241; 95% CI [0.916–5.486]; P = 0.077), elevated CK (OR = 3.137; 95% CI [1.521–6.467]; P = 0.002), refractory itching (OR = 3.267; 95% CI [1.274–8.378]; P = 0.014), above 50 years of age (OR = 2.200; 95% CI [1.060–4.568]; P = 0.034) and elevated CA19-9 (OR =2.672; 95% CI [1.034–6.902]; P = 0.042) were more likely to develop malignancies. While patients with ILD (OR = 0.440; 95% CI [0.207–0.936]; P = 0.033) were less likely to have malignancies.

Table 2 Univariate and multivariate analysis of risk factors in training cohort.

Factors	Univariate		Multivariate	
	P	OR (95% CI)		P	OR (95% CI)	
Age ≥50y	0.034a	2.200 (1.060, 4.568)		0.007*	3.534 (1.414, 8.831)	
Sex	0.084a	1.864 (0.909, 3.780)		0.180	1.804 (0.761, 4.276)	
Gottron’s sign	0.286	0.682 (0.337, 1.379)				
Periochia erythema	0.888	1.081 (0.365, 3.205)				
Poikiloderma	0.077a	2.241 (0.916, 5.486)		0.051	2.890 (0.994, 8.407)	
Refractory itching	0.014a	3.267 (1.274, 8.378)		0.013*	4.642 (1.375, 15.671)	
V-neck sign	0.857	1.067 (0.527, 2.163)				
Periorbital erythema	0.507	1.295 (0.603, 2.781)				
Raynaud’s phenomenon	>0.999	<0.001				
Joint pain	0.165	0.408 (0.115, 1.449)				
Proximal muscle weakness	0.769	0.892 (0.417, 1.908)				
Dysphagia	<0.001a	4.224 (1.932, 9.233)		0.003*	4.223 (1.617, 11.023)	
Interstitial pneumonia	0.033a	0.440 (0.207, 0.936)		0.031*	0.367 (0.147, 0.913)	
Respiration failure	0.250	0.546 (0.195, 1.531)				
CK ≥198U/L	0.002a	3.137 (1.521, 6.467)		0.049*	2.306 (1.003, 5.304)	
LDH ≥300U/L	0.784	0.905 (0.443, 1.847)				
ANA (+)	0.316	0.696 (0.343, 1.412)				
Anti-Jo-1(+)	>0.999	<0.001				
CA125 ≥35 U/ml	0.876	1.100 (0.331, 3.660)				
CA19-9 ≥37 U/ml	0.042a	2.672 (1.034, 6.902)		0.319	1.811 (0.563, 5.824)	
Notes.

OR odds ratio

CI confidence interval

a P < 0.1.

* P < 0.05, statistically significant difference.

After further multivariate analysis (Table 2), we found that the male gender, poikiloderma and elevated CA19-9 could not independently predict malignancy in dermatomyositis patients. For the remaining predictors, patients above 50 yeard old (OR = 3.534; 95% CI [1.414–8.831]; P = 0.007), refractory itching (OR = 4.642; 95% CI [1.375–15.671]; P = 0.013), dysphagia (OR = 4.223; 95% CI [1.617–11.023]; P = 0.003), and elevated CK (OR = 2.306; 95% CI [1.003–5.304]; P = 0.049) were risk factors for dermatomyositis patients with malignancy. On the other hand, ILD (OR = 0.367; 95% CI [0.147–0.913]; P = 0.031) was a protective factor.

Development and validation of the predictive nomogram

Based on the predictive factors selected by the univariate and multivariate logistic regression, a nomogram was constructed to predict the probability of malignancy occurrence in patients with dermatomyositis (Fig. 2). The value of each factor can be determined based on the intersection of the vertical line drawn from the variable to the point axis, and then the total risk score was calculated by adding all the variable points. The probability of malignant occurrence can be read directly on the total point axis. In addition, to facilitate the calculation of the probability of dermatomyositis with underlying malignancy, we provided the corresponding table form of the nomogram prediction model (Table S2), which provided the risk points of each factor and the corresponding probability to the total risk points.

Figure 2 Nomogram for individualized prediction of malignancy in patients with dermatomyositis.

In the training cohort, the C-index of the nomogram was 0.780 (95% CI [0.690–0.870]). The calibration curve showed a good consistency between predictions from the nomogram and the actual observations (Fig. 3A). The performance of the nomogram was also evaluated in the validation cohort, with a C-index of 0.756 (95% CI [0.618–0.893]) (Fig. 3B). The calibration curve also showed that the probability predicted by the nomogram was highly compatible clinically. In addition, the calculated AUC was 0.756 (95% CI [0.600–0.833]), indicating that the model has a good discrimination ability (Fig. 4).

Figure 3 Calibration curve comparing predicted and actual probabilities of dermatomyositis with malignancy in the training cohort (A) and in the validation cohort (B).

Figure 4 Receiver under the operator characteristic (ROC) curve for the test accuracy in the validation cohort.

Discussion

Dermatomyositis with underlying malignancy has a high incidence and appears concurrently or successively. Malignancy is one of the triggering factors for dermatomyositis, which induces autoimmune mechanisms through MSAs. It has been determined that certain MSAs are significantly associated with tumor-related dermatomyositis, such as TIF-1γ, NXP-2, SAE and HMGCR. Malignancy induces the expression and secretion of abnormal proteins (tumor-specific antigens), which are highly immunogenic and strongly cross-react with normal tissues. In addition, malignancy can express normal host proteins (tumor-associated antigen, TAA). Because TAAs are not specific to tumor cells, loss of tolerance to TAAs may cause the host to develop immunity against normal tissues, resulting in autoimmunity (Adler & Christopher-Stine, 2018). The sequence of dermatomyositis immune activation is unknown and may result from excessive complement activation (De Wane, Waldman & Lu, 2020). Early diagnosis of malignancy significantly impacts prognosis, especially for tumor types that lack specific indicators and are difficult to diagnose early on. Dermatomyositis is considered to be a paraneoplastic syndrome, and most dermatomyositis patients have no other specific manifestations upon diagnosis. However, malignancy is not the only triggering factor for patients with dermatomyositis. Therefore, screening malignant patients with a high risk of dermatomyositis is of great significance for improving the early detection rate of tumors and improving the tumor prognosis.

To guide clinical management strategies, this study established a risk prediction model for dermatomyositis patients with underlying malignancy and screened out five predictors, namely age, dysphagia, refractory itching, ILD and elevated CK. Dysphagia, refractory itching, and elevated CK accounted for the higher scores in the predictive model and were significantly associated with malignancy. These symptoms and the results of laboratory tests were also related to dermatomyositis activity, indicating that immuno-inflammatory response was upregulated in vivo, and the autoimmune antibody titers were increased, leading to tumor-associated immune pathways activation. Besides, age also had a higher score in the prediction model. Almost all previous studies considered age to be a clear predictor, which is consistent with the biological behavior of tumors. Consequently, it is recommended that dermatomyositis patients over the age of 50 routinely undergo screening for tumors, including tumor-related history, physical examination, age-appropriate tumor marker screening, and whole-body imaging (CT, MRI or PET-CT) screening and even biopsy of suspicious lesions. ILD was found to be a protective factor for dermatomyositis associated with malignancy. Dermatomyositis patients with concurrent ILD had a significantly lower risk of developing malignancy, which is in line with the results of other highly credible clinical studies (Lu et al., 2014; Wang et al., 2013; Lega et al., 2014; Ikeda et al., 2015). ILD is significantly related to the anti-MDA5 antibody (melanoma differentiation-associated gene 5, MDA5), especially aggressive ILD with a poor prognosis. Compared with typical dermatomyositis patients, the degree of IFN-related pathways activation in MDA5 positive patients was lighter, accompanied by a lower-activating state of STAT1, leading to a down-regulation of NOS2 expression in Th1 macrophages. At the same time, it was observed that MDA5 positive dermatomyositis patients specifically expressed NOS2 in sarcoplasmic muscle fibers and co-localized with the protective HP70 chaperone protein. These mechanisms suggest that ILD might play a role in tumor suppression (Allenbach et al., 2016). However, the current negative correlation between ILD and tumors in patients with dermatomyositis warrants further exploration.

In previous studies, a meta-analysis (Lu et al., 2014) comprehensively evaluated the risk factors and protective factors of dermatomyositis associated with malignancy. There were several similarities and differences between the conclusions of this article and previous ones. Herein, we reported that age, dysphagia, and refractory itching are risk predictors, in line with the previous meta-analysis’s conclusion. However, gender was not considered a predictor in this study; although previous studies found the opposite. The meta-analysis had a lower OR value for gender as a risk factor (OR = 1.92, 95% CI [1.49–2.48]), indicating that the reliability of gender as a high-risk factor was uncertain. The reliability of gender in epidemiology depends mainly on the number of cases, and thus, increasing the number of cases may change the status of the gender in the risk prediction model. ESR, CRP, WBC and other laboratory tests were not considered predictors, which might be due to the close relationship between laboratory tests levels and the patient’s current inflammation and immune status that tend to have large fluctuations throughout the course of the disease. A retrospective study has shown that heliotrope rash and Gottron’s sign are also risk factors for malignancy in dermatomyositis patients (Targoff et al., 2006), which might be related to positive anti-p155 antibody level. Anti-p155 positive dermatomyositis patients have an increased risk of malignancy (Trallero-Araguas et al., 2012) and often have poikiloderma, Gottron’s sign, heliotrope rash, and V-neck sign (Targoff et al., 2006). However, our findings show that these vasculitis-related lesions are not related to the onset of malignancy in patients with dermatomyositis, which might be due to the insufficient number of cases that cannot reflect the statistical differences of these variables.

A nomogram clinical prediction model for dermatomyositis patients with underlying malignancy was established for the first time based on five predictors. The prevalence of dermatomyositis is low, about 7-9/1,000,000 (Sunderkotter et al., 2016), which greatly limited the number of cases recruited for investigation. However, the internal and external calibration curves showed consistency with clinical observations and had good predictive power. Our external verification did not use participants from other institutions; instead, new participant data at a later period in the same database was used. This verification method has been used before and conforms to the norm (Moons et al., 2015). More importantly, the predictive indicators of this model are easy to obtain and observe, which is beneficial to physicians and has great value for identifying malignancy in patients with dermatomyositis.

Nevertheless, this study has several limitations: (1) The number of cases investigated was small since it was a single-center study. (2) The patients in the study were all from southern China. Southern China is in the subtropical zone, closer to the equator, and is different from other regions in terms of climate and environment. Some studies suggested that multiple environmental factors can also trigger chronic immune activation in patients with genetic susceptibilities, such as ultraviolet light and viral infections (Love et al., 2009). Therefore, this study might be representative of the patient population in the south of China only. (3) Cases with incomplete clinical data have been excluded leading to possible publication bias. (4) The median follow-up length of patients with dermatomyositis was insufficient, which might lead to incomplete data acquisition since some patients could have developed malignant tumors at a later stage of the disease. In the future, prospective studies should be carried out with long-term follow-ups of dermatomyositis patients to obtain a more accurate incidence rate of malignant tumors.

Conclusions

This study established a simple, intuitive, and practical nomogram risk prediction model for dermatomyositis patients with underlying malignancy. Following internal and external validation, the model demonstrated a good calibration and discrimination ability. Altogether, our nomogram has good clinical translational potential, possibly helping physicians identify dermatomyositis patients at higher risk of developing malignancy. This could ultimately lead to early tumor-related screening and follow-up of high-risk patients to improve their clinical outcome considerably. However, in order to optimize the model and improve the accuracy of prediction model, it is still necessary to collect cases from multiple centers and different regions.

Supplemental Information

Supplemental Information 1 Type of cancer in dermatomyositis patients with malignant tumors

Click here for additional data file.

Supplemental Information 2 Points of risk factors and risk of total points

Click here for additional data file.

Supplemental Information 3 Raw Data

Clinical and laboratory data on dermatomyositis patients in the training cohort and the validation cohort. The data were used for statistical analysis.

Click here for additional data file.

Supplemental Information 4 Codebook for rawdata

Click here for additional data file.

Supplemental Information 5 TRIPOD Checklist: Prediction Model Development and Validation

Click here for additional data file.

Thanks for all patients who provided clinical data for this article.

Additional Information and Declarations

Competing Interests

Author Contributions

Human Ethics

Data Availability

The authors declare that they have no competing interests.

Jiaojiao Zhong and Yunan He conceived and designed the experiments, performed the experiments, analyzed the data, prepared figures and/or tables, authored or reviewed drafts of the paper, and approved the final draft.

Jianchi Ma and Junmin Zhang conceived and designed the experiments, authored or reviewed drafts of the paper, and approved the final draft.

Siyao Lu performed the experiments, analyzed the data, prepared figures and/or tables, and approved the final draft.

Yushi Wu performed the experiments, prepared figures and/or tables, and approved the final draft.

The following information was supplied relating to ethical approvals (i.e., approving body and any reference numbers):

This clinical study was reviewed and approved by the Ethics Committee of Sun Yat-sen Memorial Hospital, Sun Yat-sen University (SYSEC-KY-KS-2020-038).

The following information was supplied regarding data availability:

The raw measurements are available in the Supplementary File.

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
