# Peer review of "Development and validation of a nomogram risk prediction model for malignancy in dermatomyositis patients: a retrospective study"

_PeerJ, doi:10.7717/peerj.12626_

## Round 0.1 · original submission · Minor Revisions

Please follow strictly the suggestions and concerns of the reviewers.

Reviewer 1 ·

Basic reporting

In my opinion, it is a very well written, novel, relevant paper. Introduction section provides sufficient details although I suggest expanding information on nomogram in the last paragraph, and briefly describe if nomogram has been used for this condition before, if yes, what were the problems? How this study is better?
In discussion section, authors may want to comment on the short median length of follow up, if this is another limitation, also will it be to conduct a prospective study?

Experimental design

This part is also well written, within the aims and scope of this journal. the research question has been clearly defined and its relevant to the field. Methods are robust and meet the ethical standards. Also, there is sufficient detail to replicate the experiments.

I have a minor comment here, would it be possible to elaborate on was there any history of smoking in patients with lung cancer or any other collateral history that may be a risk factor for respective malignancy?

Validity of the findings

The underlying data have been provided, which proves study to be novel, statistically sound, and controlled.
Conclusions address the original research question and support results.

Annotated reviews are not available for download in order to protect the identity of reviewers who chose to remain anonymous.

Reviewer 2 ·

Basic reporting

Recommend hiring an English language service to assist with academic prose. The term "periochia" is not a generally accepted term in the English literature - recommend use of the term "periungual" instead.

Background, structure, raw data, hypotheses are all acceptable.

Experimental design

Research question well defined and investigation performed to technical and ethical standards. Methods sufficiently described.

Validity of the findings

Use of CXR alone may be underdiagnosing ILD in DM patients – could have much higher rates among both cohorts (malignancy-associated and non-malignancy-associated) which could affect model.

Usually build multivariable analysis from univariable analyses where p<0.1 rather than p<0.05; a variable which may not be important in univariable analysis may occasionally become important in multivariable analyses, thus larger p-values than the traditional p<0.05 are typically used. Correcting this in univariable analysis could introduce factors such as sex (as has been found in previous studies) and poikiloderma, both of which have 0.05<p<0.1.

Additional comments

In addition to its current form, would consider an alternative figure/table depicting the points assigned to each criterion in the nomogram, so the point total can be more easily calculated without the need for a straightedge.

---

## Round 0.2 · Minor Revisions

Recommend English language service for assistance with grammar. There are instances in the introduction (lines 40-42, 52-54, 59-61, 64-68), methods (lines 84-85, 93-94, 104 [the plural of index is indices]), and results (lines 134-135,153-154) that do not use acceptable English grammar.
Please attach a VALID English certificate on this manuscript!

Reviewer 1 ·

Basic reporting

Authors have provided answer to my previous question in a satisfactory manner.

Experimental design

Methods are robust and meet the ethical standards. Also, there is sufficient detail to replicate the experiments. In addition. authors have addressed the issues raised in previous review.

Validity of the findings

The underlying data supports notion that study is novel, statistically sound, and controlled.
Conclusions support results.

Additional comments

From my perspective, study has been improved substantially and its suitable for publication.

Reviewer 2 ·

Basic reporting

Recommend English language service for assistance with grammar. There are instances in the introduction (lines 40-42, 52-54, 59-61, 64-68), methods (lines 84-85, 93-94, 104 [the plural of index is indices]), and results (lines 134-135,153-154) that do not use acceptable English grammar.

Added background is appreciated and adds to the manuscript.

Experimental design

Appreciate corrections to statistical modeling methodology.

Validity of the findings

Conclusions are well-stated and supported by statistical analysis.

Additional comments

Appreciate the hard work the authors put into making revisions to this manuscript. However there are a few grammatical issues that should be addressed prior to publication.

---

## Round 0.3 · accepted · Accept

Thank you for your submission!

Reviewer 1 ·

Basic reporting

The current version is a substantially improved version of the original manuscript.

Experimental design

Methods are well reported with sufficient amount of details.

Validity of the findings

All underlying data has been provided and conclusion supports the results.

Additional comments

In my opinion, in the current revised version, a substantial amount of improvement can be noticed. Authors seems to have work hard to make all the necessary changes. Writing style is much better and it facilitates reading.

Reviewer 2 ·

Basic reporting

Appreciate use of an English language service and corrections to grammar. No other concerns.

Experimental design

No additional comments

Validity of the findings

No additional comments